# Optimization and Characterization of a Novel Exopolysaccharide from *Bacillus haynesii* CamB6 for Food Applications

**DOI:** 10.3390/biom12060834

**Published:** 2022-06-15

**Authors:** Aparna Banerjee, Sura Jasem Mohammed Breig, Aleydis Gómez, Ignacio Sánchez-Arévalo, Patricio González-Faune, Shrabana Sarkar, Rajib Bandopadhyay, Sugunakar Vuree, Jorge Cornejo, Jaime Tapia, Gaston Bravo, Gustavo Cabrera-Barjas

**Affiliations:** 1Centro de Investigación de Estudios Avanzados del Maule, Vicerrectoría de Investigación y Posgrado, Universidad Católica del Maule, Talca 3466706, Chile; runka.sarkar@gmail.com; 2Centro de Biotecnología de los Recursos Naturales (CENBio), Facultad de Ciencias Agrarias y Forestales, Universidad Católica del Maule, Talca 3466706, Chile; aleydisgomez1965@gmail.com; 3Department of Biochemical Engineering, Al-khawarizmi Collage of Engineering, University of Baghdad, Baghdad 10011, Iraq; sura.breig@yahoo.com; 4Escuela de Ingeniería en Biotecnología, Facultad de Ciencias Agrarias y Forestales, Universidad Católica del Maule, Talca 3466706, Chile; ignacio.sanchez@alu.ucm.cl (I.S.-A.); patricio.gonzalez@alu.ucm.cl (P.G.-F.); 5UGC Center of Advanced Study, Department of Botany, The University of Burdwan, Burdwan 713104, India; rajibindia@gmail.com; 6Department of Biotechnology, School of Bioengineering and Biosciences, Lovely Professional University, Phagwara 144411, India; sugunakar.24344@lpu.co.in; 7Institute of Chemistry and Natural Resources, Universidad de Talca, Talca 3460000, Chile; jocornejo@utalca.cl (J.C.); jtapia@utalca.cl (J.T.); 8Unidad de Desarrollo Tecnológico (UDT), Universidad de Concepción, Av. Cordillera 2634, Parque Industrial Coronel, Coronel 3349001, Chile; g.bravo@udt.cl

**Keywords:** *Bacillus*, exopolysaccharide, food industry, additive, antioxidants, emulsification, flocculation, water holding, oil holding

## Abstract

Extremophilic microorganisms often produce novel bioactive compounds to survive under harsh environmental conditions. Exopolysaccharides (EPSs), a constitutive part of bacterial biofilm, are functional biopolymers that act as a protecting sheath to the extremophilic bacteria and are of high industrial value. In this study, we elucidate a new EPS produced by thermophilic *Bacillus haynesii* CamB6 from a slightly acidic (pH 5.82) Campanario hot spring (56.4 °C) located in the Central Andean Mountains of Chile. Physicochemical properties of the EPS were characterized by different techniques: Scanning electron microscopy- energy dispersive X-ray spectroscopy (SEM-EDS), Atomic Force Microscopy (AFM), High-Performance Liquid Chromatography (HPLC), Gel permeation chromatography (GPC), Fourier Transform Infrared Spectroscopy (FTIR), 1D and 2D Nuclear Magnetic Resonance (NMR), and Thermogravimetric analysis (TGA). The EPS demonstrated amorphous surface roughness composed of evenly distributed macromolecular lumps. GPC and HPLC analysis showed that the EPS is a low molecular weight heteropolymer composed of mannose (66%), glucose (20%), and galactose (14%). FTIR analysis demonstrated the polysaccharide nature (–OH groups, Acetyl groups, and pyranosic ring structure) and the presence of different glycosidic linkages among sugar residues, which was further confirmed by NMR spectroscopic analyses. Moreover, D-mannose α-(1→2) and α-(1→4) linkages prevail in the CamB6 EPS structure. TGA revealed the high thermal stability (240 °C) of the polysaccharide. The functional properties of the EPS were evaluated for food industry applications, specifically as an antioxidant and for its emulsification, water-holding (WHC), oil-holding (OHC), and flocculation capacities. The results suggest that the study EPS can be a useful additive for the food-processing industry.

## 1. Introduction

Food industries around the world are searching for value-added compounds or additives of natural origin with increased functionality and bioactivity. Moreover, in the present time, an increasing trend may be observed in the consumer toward healthier foodstuffs [1]. Bacterial exopolysaccharides (EPSs) play an important role in the improvement of the rheological and sensory characteristics of food products by positively influencing the food texture and organoleptic properties. Additionally, EPSs have also gained potential research interest for pharmacological and nutraceutical applications due to their biocompatibility, non-toxicity, and biodegradability [2]. Lactic acid bacteria are already well known for their techno-functional food applications in the dairy industry [3]. Additionally, EPSs produced by bacteria from extreme environments are also well reported for applications in the food industry, for example, as antioxidants, stabilizing emulsions, pseudoplastic behavior, and more [4]. Food industry processes are generally exposed to extreme temperature, pH, and salinity [5]. Extremophiles are known for their wide range of adaptability in extreme conditions and are additionally stable over a large range of temperature, pH, and salinity [4].

Extreme environmental conditions with more than one extreme factor, i.e., polyextreme environments, are gaining much scientific interest because of their dynamic community interactions, diversity, and adaptive strategies [6]. In this context, the unique geographical location of Chile provides a natural laboratory to study microbial community dynamics in polyextreme environments [7]. Bacterial communities can withstand extremes for survival, including extreme pH, temperature, salinity, pressure, drought, UV irradiation, heavy metals, and toxic compounds [8]. Bacterial physiology and adaptations in harsh environments are influenced by the EPS they produce. EPSs are polydisperse, polyfunctional, and often poorly structurally/functionally characterized [9]. EPSs produced by the thermophiles are reported to have good thermal stability and several biotechnological properties. They are capable of stabilizing other molecules, show good synergy with hydrocolloids, and have antimicrobial, and immunostimulant properties determining their possible future applications [10]. Bacterial EPSs have been considered promising antioxidants to develop effective functional foods with a longer shelf life [11]. According to a previous report, the EPS produced by thermophilic *Geobacillus* sp. showed 100% emulsification of different edible food-grade oils [12], which indicates their potential as natural emulsifiers replacing synthetic emulsifying agents. In one of our previous studies, thermotolerant *Bacillus* species from an Indian hot spring produced an EPS with a higher antioxidant capacity than commercial ascorbic acid [13]. On the other side, EPSs have flocculating activity due to functional groups on their polymeric backbone [14]. Microbial-origin flocculants are regularly used in the food and feed industry for non-toxic, residue-free biomass harvesting [15]. Furthermore, EPSs with water-holding and oil-holding capacities are also exploited to prepare functional food products by stabilizing water and oil emulsions [16,17]. According to Wang et al. [4], the present trend in extremophilic EPS research has the main constraint of production cost, which is a barrier to their replacing commercial EPS biopolymers. Thus, the production and characterization of extremophilic bacterial EPS is an active research field due to the continuous need to improve cultivation techniques, economically.

Our present work reveals the biotic potential of an unexplored hot spring located in central Chile that is volcanic in origin, slightly acidic, rich in heavy metals, and harboring possibilities of novel bioactive compounds produced by the thermophilic bacterial community. In this study, a new EPS from the thermophilic *Bacillus haynesii* is reported, EPS production is optimized, and the optimized condition is statistically validated. The EPS is morphologically, structurally, and functionally elucidated as a novel antioxidant, emulsifier, and flocculating agent, showing promising biotechnological applications for the future. According to our knowledge, this is the first report of any characterized bacterial EPS from Chilean hot springs. 

## 2. Materials and Methods

### 2.1. Sampling and Water Analysis

In this study, water samples were collected at a depth of almost 0.5 m from the Campanario hot spring, located in the central Andean Mountains of Chile in the Maule region (35°56′23″ S 70°36′22″ W). The temperature, pH, and conductivity of the water were measured on-site using a pH/ORP meter (LAQUA PH120-K, HORIBA, Kyoto, Japan). For the detailed physicochemical analysis of the collected water, samples were filtered in polycarbonate systems, using filters of 0.45 µm porosity. The analysis was performed according to the Standard Methods for the Examination of Water and Wastewater as described by INN-Chile [18,19,20]. All the reagents used were of high purity (Suprapure, Merck, Darmstadt, Germany). In the present study, detailed physicochemical parameters of collected hot-spring water were determined, as follows: dissolved oxygen (DO), biochemical oxygen demand (BOD), total dissolved solids (TDS), total alkalinity, chlorides, color, turbidity, sulfates, nitrates, and sulfates. Fixation was carried out to determine the concentration of dissolved oxygen according to the Winkler method (Standard Methods for the Examination of Water and Wastewater). Taking into consideration the Laguna del Maule volcanic field where the hot spring is located, the presence of different heavy metals in the water sample was detected, viz., aluminum (Al), arsenic (As), cadmium (Cd), copper (Cu), chromium (Cr), iron (Fe), zinc (Zn), manganese (Mn), magnesium (Mg), mercury (Hg), nickel (Ni), and lead (Pb). To determine the metals (except Hg and As) in the water, a 500 mL-filtered sample was pre-concentrated by the addition of 1 mL of 70% nitric acid. Pre-concentration was carried out by evaporation using a Thermolyne Cimarec hot plate at 105 °C to a final volume of 50 mL. Metal concentration was determined by atomic absorption spectroscopy (AAS) (Perkin Elmer 1990, Norwalk, Conn, USA) using a spectrophotometer (Thermo Fisher Scientific ICE 3000 Series, Cambridge, UK) as previously mentioned by Fierro et al. [21]. The determination of Hg was carried out using the cold vapor technique, and As by hybrid evolution. The detection limits for metals were as follows: Al 28.0 µg L^−1^; As 0.04 µg L^−1^; Cd 2.8 µg L^−1^; Cu 4.5 µg L^−1^; Cr 5.4 µg L^−1^; Fe 4.3 µg L^−1^; Mn 1.6 µg L^−1^; Mg 2.2 µg L^−1^; Hg 0.06 µg L^−1^; Ni 8.0 µg L^−1^; Pb 13.0 µg L^−1^ and Zn 3.3 µg L^−1^.

### 2.2. Isolation and Taxonomic Identification of the Bacteria

Serial dilution of the collected water sample was performed and 50 μL of each dilution was inoculated aseptically in nutrient agar (NA) (Difco) media (pH 5.80). As the surface water temperature of the hot spring was observed to be 56.4 °C, for synchronization of bacterial growth, the inoculated media was incubated at 55 °C. After 72 h, a white, mucoid colony of isolate CamB6 appeared, which was further chosen for EPS production. Taxonomic identification of the mucoid isolate was recently reported in Marín-Sanhueza et al. [22], where bacteria were optimally grown at 55 °C, in nutrient broth (NB) media (Difco). Briefly, DNA was isolated and the 16S rRNA region was amplified with the universal 27F and 1492R primers. The obtained amplicons were sequenced and EzTaxon was used for identification (https://www.ezbiocloud.net/identify; Accessed 10 January 2021). Evolutionary distances between the sequences were calculated [23] and the phylogenetic tree was prepared following the maximum likelihood method using MEGA X [24], followed by deposition in GenBank.

### 2.3. Recovery, Purification of EPS, and One Factor at a Time (OFAT) Optimization 

The method for EPS recovery was followed as described by Rimada and Abraham [25]. For the production of EPS, stationary phase bacterial cells, harvested in modified NB, were treated with 4% trichloroacetic acid (*w/v*) for 30 min. Next, TCA-precipitated cells were centrifuged at 4 °C, 5000× *g* for 20 min to precipitate the proteins. An equal volume of chilled acetone was added to the chilled cell-free supernatant and left overnight at 4 °C. The solvent-coagulated EPS was then separated by centrifuging the solution at 4 °C and 12,000× *g* for 20 min. The protein-purified EPS was then dialyzed, followed by lyophilization to obtain the pure EPS powder. Initially, classical one-factor-at-a-time (OFAT) optimization was applied, altering one factor of EPS production, while the other factors were fixed at a certain value [26]. OFAT optimization was used to screen the carbon and nitrogen sources that augment EPS production by isolate CamB6. Different carbon sources (xylose, mannose, glucose, and sucrose) and nitrogen sources (yeast extract, casein hydrolysate, and ammonium sulfate) were optimized for this experiment. Each experiment was conducted in triplicate to stabilize the experimental error. 

### 2.4. Response Surface Methodology (RSM) Optimization and Its Validation

In this study, RSM based on central composite design (CCD) [27] was used to optimize the EPS production for four parameters. These were carbon source (glucose) 20–40 g L^−1^, nitrogen source (yeast extract) 20–40 g L^−1^, pH 6–8, and inoculum size 20–40 mL L^−1^. Five levels of each parameter were coded as (-α, −1, 0, 1, +α) and used as independent parameters with EPS production as a response. Table 1 exhibits coded and uncoded values of independent parameters, using Equation (1): x_i_ = (X_i_ − X_o_/ΔX)(1)
where: x_i_ is the coded value of the parameter, X_i_ is the actual value of the parameter, X_o_ is the actual value of X_i_ at the center point, and ΔX is the step change.

The CCD matrix was generated by Design Expert software version 7 (StatEase, Minneapolis, USA) with thirty runs and six replications of the center point, as shown in Table 2. Design Expert is a beneficial tool commonly used for experiment design and response data analysis [28]. All the experiments were performed in triplicate. The inoculated flasks were incubated at 55 °C and 150 rpm for 3 days in a rotary shaker to harvest the stationary phase bacterial cells. After production, the EPS concentration was detected and the achieved data were analyzed to evaluate the effect of each parameter and their interaction to predict the optimized EPS production, as described in the quadric optimization model (Equation (2)). (2)$$Y=\beta_{0}+\sum_{i=1}^{k}\beta_{i}x_{i}+\sum_{i=1}^{k}\beta_{\mathrm{ii}}x_{i}^{2}+\sum_{i=1}^{k-1}\sum_{i=2}^{k}\beta_{\mathrm{ij}}x_{i}x_{j}$$
where Y is the predicted response, β_0_ is the intercept term, β_i_ is the linear effect, β_ii_ is the squared effect, β_ij_ is the interaction effect, and X_i_ and X_j_ are input variables that influence the response variable Y.

The optimization model was statistically evaluated to assess the analysis of variance. Model fitness was expressed by R^2^, R adjust, R predict, adequate precision, standard deviation, coefficient of variation, mean, press, F-value, P > F prob, and lack of fit [28]. Depending on the regression model, optimization plots were developed using Design Expert to identify the optimum concentrations of all the four studied parameters. For the maximized EPS production, all independent parameters were selected in range, and response EPS was maximized to construct a ramp chart to describe the optimization. Optimal parameters were validated in the laboratory to verify the predicted EPS production. 

### 2.5. Physicochemical Characterization of the EPS

#### 2.5.1. Morphological Analysis (AFM and SEM)

The surface morphology of macromolecules helps to identify their common physical properties [29]. Atomic force microscopy (AFM) and scanning electron microscopy (SEM) were employed to study the surface morphology of the EPS. For the AFM analysis, a 0.01 mg mL^−1^ aqueous EPS solution was prepared. The solution was vortexed well at room temperature to ensure complete dissolution. Approximately 5 μL of the EPS solution was dropped onto the cover slip and air-dried. Finally, the surface images of the EPS were obtained using an AFM (diINNOVA, Bruker, Madison, WI, USA) at the tapping mode of the NanoDrive Innova system. 

For SEM analysis, the EPS was made conductive by gold-coating, using a Gold Sputter Coater (DII 29030SCTR Smart Coater, JEOL, Tokyo, Japan), and its surface structure was observed under FESEM (JEOl-JSM 7610FPlus) at 15 kV accelerating voltage. The elemental mapping of the EPS was performed using energy dispersive X-ray spectroscopy (EDS) (X-Max-AZtec, Oxford Instruments) for analysis of the carbon/nitrogen/oxygen/phosphorus/sulfur composition. 

#### 2.5.2. Monosaccharide Composition Analysis and Molecular Weight

A high-pressure liquid chromatography (HPLC) system (Shimadzu, Kioto, Japan) was equipped with an LC-20AT Pump, multiple autosamplers (SIL-20A) fitted with a 20-µL loop, a UV detector (SPD-20AV) set to 210 and 290 nm, and a refractive index detector (RID-10A). The detectors were connected in series. Data were collected on a LabSolution software 1.25 Version system (Shimadzu, Kyoto, Japan). The analyses were performed isocratically with water at 0.8 mL min^−1^ and 65 °C with a 300 × 7.8 mm I.D. cation exchange column (Aminex HPX-87H) equipped with a cation H^+^ microguard cartridge (Bio-Rad Laboratories, Hercules, CA, USA). Sugar standards were supplied by Sigma Chile. Different calibration curves using glucose, galactose, and mannose standards were built for monosaccharides quantification. The sugar concentration ranged from 200 to 10,000 µg mL^−1^. The calibration parameters obtained for each curve were glucose (140x − 6277, R^2^ 0.999), galactose (101x − 9566, R^2^ 0.999) and mannose (147x – 26,559, R^2^ 0.997).

Gel permeation chromatography (GPC) allows determination of the average molecular weight (Mw) distribution of the isolate CamB6-produced EPS, using an 1100 HPLC system (Agilent Technologies, Santa Clara, USA), attached onto Shodex columns (40, 300, 1000). Due to the polysaccharide nature of the EPS, Pullulan (Sigma, St. Louis, United States) standards (0.3–700 kDa) were used for GPC analysis. The experiments were conducted under the following conditions: samples 10 μL, 0.1 g mL^−1^; solvent 0.1 M sodium nitrate; Φ = 0.5 mL min^−1^; temperature = 50 °C. 

#### 2.5.3. Structural Analysis 

##### Infrared Spectroscopy 

Fourier transform infrared (FTIR) spectrum for the EPS sample was acquired in transmittance mode with an FTIR-ATR spectrometer (Jasco-4000, Jasco Analytical Spain, Madrid, Spain) to analyze different functional groups. The spectrums were recorded by pressing the samples into KBr pellets at a 1:90 ratio; these were then scanned in the range of 4000–500 cm^−1^ with a resolution of 4 cm^−1^.

##### 1D and 2D NMR Analysis

NMR analyses were performed at 60 °C after samples had been dissolved in D**_2_**O. Chemical shifts of water-soluble samples are expressed in d ppm relative to DSS as an internal reference for 13C and 1H signals, respectively. 1D (1H and 13C) and 2D NMR spectra (1H (obsd.) 13C, COSY, HSQC) were obtained by following the Bruker manual. 1H NMR and 13C NMR spectra were recorded at 400 and 100 MHz. Chemical shifts (δ) are reported in parts per million (ppm) relative to the residual solvent signals. (Appendix A).

#### 2.5.4. Thermogravimetric Analysis

The thermal stability of the EPS was determined using a thermogravimetric analyzer (TGA) Cahn-Ventron 2000 (Cahn Scientific, Irvine, CA, USA) with a microprocessor-driven temperature-control unit and a thermal analysis data station. Approximately 5 mg of sample was placed in an aluminum sample pan with a temperature raising range from 25 to 600 °C, at a heating rate of 10 °C min^−1^ under an N_2_ gas flow of 50 mL min^−1^.

### 2.6. Functional Properties of the EPS

#### 2.6.1. In Vitro Antioxidant Activity Determination

The 2,2-diphenyl-1-picrylhydrazyl (DPPH) radical scavenging activity (RSA) was assayed according to Shimada et al. [30] with modifications. Briefly, 200 μL of EPS aqueous aliquot (0.2, 0.5, 1, 2 and 5 mg mL^−1^) was added to 0.2 mM 20 μL ethanolic DPPH solution. After vigorous mixing, it was incubated in the dark at 30 °C for 1 h. After centrifugation at 5000× *g* (10 min), absorbance of the supernatant was recorded at 517 nm using a Mobi-Microplate Spectrophotometer (μ2 MicroDigital, MOBI, Seoul, South Korea). DPPH RSA was calculated using Equation (3): [(A_0_ − A_1_)/A_0_] × 100 (3)

where A_0_ is the absorbance value of DPPH solution without the sample and A_1_ is the absorbance value of the EPS solution. Ascorbic acid was used as the positive control.

Free radicals derived from 2,2′-azino-bis (3-ethylbenzothiazolin-6-sulfonic acid)/ABTS and its scavenging activity were evaluated following the method used by Nitha et al. [31]. The ABTS solution was diluted to an absorbance of ~0.75 at 734 nm in phosphate buffer (pH 7.40). Then, 180 μL of different EPS concentrations (0.2, 0.5, 1, 2, and 5 mg mL^−1^) were added to 20 μL of ABTS radical solution and allowed to react for 5 minutes at 30 °C. Absorbance was recorded at 734 nm using quercetin as a reference in the Mobi-Microplate Spectrophotometer (μ2 MicroDigital). The scavenging activity was calculated using Equation (3). In this case, A_0_ is the absorbance value of ABTS radical solution without the sample and A_1_ is the absorbance value of EPS solution. Ascorbic acid was used as the positive control.

The H_2_O_2_ scavenging activity was determined according to Ruch et al. [32] with some modifications. The mixture containing 50 μL of the sample (0.2, 0.5, 1, 2, and 5 mg mL^−1^), 120 μL of phosphate buffer (0.1 M, pH 7.40), and 30 μL of H_2_O_2_ solution (40 mM) was shaken vigorously and incubated at 30 °C for 10 minutes. The absorbance of the reaction mixture was then recorded at 230 nm using the Mobi-Microplate Spectrophotometer (μ2 MicroDigital). The H_2_O_2_ scavenging activity was calculated as follows (Equation (4)): [1 − (A_1_ − A_2_)/A_0_] × 100 (4)
where A_0_ is the absorbance of the control MiliQ water, A_1_ is the absorbance of the sample, and A_2_ is the absorbance of the sample without H_2_O_2_ solution. Ascorbic acid was used as the positive control.

#### 2.6.2. Emulsifying Activity Study

The emulsifying activity of the EPS produced by thermophilic CamB6 was measured using the modified method used by Cooper and Goldenberg [33]. For this, food-grade vegetable oils (coconut, corn, canola, avocado, sunflower, olive, and sesame) were added to a 1 mg mL^−1^ aqueous phase containing the EPS (oil: EPS in a ratio of 3:2, *v*/*v*) and agitated vigorously for 2 min on a vortex. The oil, emulsion, and aqueous layers were measured at 24 h intervals. To observe the emulsion stability, the emulsification index (E) was calculated as: [(volume of the emulsion layer × total volume^−1^) × 100]. As xanthan gum is one of the most widely used commercial bacterial EPS sources regularly used as an emulsifier [34], the emulsifying property of EPS was compared to other natural standard bacterial EPS xanthan gum (Sigma), along with synthetic surfactant Tween 20 (Sigma).

#### 2.6.3. Flocculation Activity Study

Flocculation activity of the EPS produced by thermophilic CamB6 was measured according to Pu et al. [35] with some modifications, along with the control xanthan gum. Briefly, 1% CaCl_2_ containing kaolin suspension (pH 7.0, 4 g L^−1^) was mixed with different concentrations of EPS (5–100 mg L^−1^) in a 1:1 (*v*/*v*) ratio, stirred well, and left for 10 minutes undisturbed. The absorbance of the supernatant was measured at 550 nm using the Mobi-Microplate Spectrophotometer (μ2 MicroDigital). The flocculating percentage was calculated by the Equation (5): [(A − B)/A] × 100 (5)
where A is absorbance of the control supernatant and B is the absorbance of the sample.

#### 2.6.4. Water-Holding and Oil-Holding Capacity Determination

The water-holding capacity (WHC) of the EPS produced by *B. haynesii* CamB6 was measured using the methods reported by Kumari et al. [36]. Briefly, 500 mg of EPS was taken in an initially weighed centrifuge tube and 10 mL of distilled water was added, followed by mixing in a cyclomixer for 1 min. The solution was kept at rest for 30 min at 37 °C with intermediate stirrings, each followed by centrifugation at 3200 rpm for 25 min. The supernatant was decanted after centrifugation and the tube was weighed. The water-holding capacity (WHC) was calculated using the following formula:
(6)$$\%\;WHC=\frac{Water\;bound\;weight\;(g)}{Initial\;sample\;weight\;(g)}\times 100$$

The oil-holding capacity (OHC) of the study EPS was calculated following the standard method used by Wang and Kinsella [37], with slight modifications. The OHC sample was prepared by mixing 500 mg of lyophilized EPS in 10 mL of sunflower oil in a cyclomixer. The solution was then allowed to stand for 30 minutes at 37 °C with intermediate shaking every 10 minutes, followed by centrifugation at 3200 rpm for 25 min. The supernatant was then decanted and the tube weighed. Oil-holding capacity (OHC) was calculated using the following formula:(7)$$\%\;OHC=\frac{Oil\;bound\;weight\;(g)}{Initial\;sample\;weight\;(g)}\times 100$$

Xanthan gum (Sigma) was used as the control for both experiments. All experiments were performed in triplicate. The experimental data were analyzed by calculating the mean ± SD and the analysis of variance (ANOVA) to determine statistically significant differences. The Bonferroni test was used, and the values were considered if *p* < 0.05. The statistical program SigmaPlot 12.0 (Systat Software Inc., London, UK)) was used for this study.

## 3. Results

### 3.1. Sampling and Water Analysis

The water samples were collected from the Campanario hot spring (35°56′23″ S 70°36′22″ W), located in the central Andean Mountains in the Maule region of Central Chile (Figure 1). In the present study, different physicochemical parameters of the collected water sample were studied: temperature, pH, conductivity, DO, BOD, TDS, total alkalinity, chlorides, color, turbidity, sulfates, nitrates, sulfates, Al, As, Cd, Cu, Cr, Fe, Mn, Mg, Hg, Ni, Pb, and Zn. The detailed physicochemical parameters can be found in Appendix A. The study site Campanario hot spring is slightly acidic (pH 5.82) and has a surface water temperature of 56.4 °C.

### 3.2. Isolation and Taxonomic Identification of EPS-Producing Thermophilic Bacteria

Of the different thermophilic bacterial colonies grown on the NA media (55 °C), a shiny white, mucoid colony CamB6 was initially isolated for its characteristic EPS production. Phylogenetic identification through EZ-Taxon analysis demonstrated that the 16S rRNA sequence of isolate CamB6 (Genbank accession no. MZ298610) showed 100% similarity with *B*. *haynesii* NRRL B-41327(T) (Accession: MRBL01000076) [38]. Other than this, 19 different *Bacillus* strains also exhibited ~97% similarity to isolate CamB6 (Figure 1).

### 3.3. Recovery, Purification of EPS, and OFAT Optimization 

The best EPS production by CamB6 was achieved at 55 °C in the stationary growth phase by the trichloroacetic acid precipitation method [39]. The best-optimized carbon source was found to be glucose, and the nitrogen source was found to be yeast extract resulting in an initial 2.9 g L^−1^ EPS production in the pre-optimized condition. This was used for further response surface methodology (RSM) analysis to statistically optimize the production.

### 3.4. RSM Optimization and Its Validation

RSM based on the central composite design technique was applied to optimize the possible combination of different studied parameters that maximize the EPS production by hot-spring-origin thermophilic *B. haynesii* CamB6. The upper and lower limit of each parameter was specified according to OFAT optimization, as mentioned earlier. The value of α was selected based on a rotatable hypothesis for constant variance at points that were equidistant from the center point and supply equal precision of response evaluation in any direction of the design [34]. The CCD matrix was constructed with 30 runs. Each showed interactions among independent parameters in one flask (Table 1). Six replications of the center point were considered to determine the experimental error, which was used to detect curvature in the response as they contribute to the estimation of the coefficient of quadratic terms. Additionally, the axial point was utilized to determine the coefficient of quadratic terms, whereas the factorial point determined the coefficients of linear terms and two-way interactions [28].

Depending on the response values of EPS production by *B. haynesii* CamB6 and data analysis from the lack of fit summary, the quadratic model was found to be the most suitable optimization model for EPS production following P > F as 0.1081. By analyzing variance (ANOVA) from the quadratic optimization model, the optimization was found to be highly significant with an F-value of 225.63 (Prob > F = < 0.0001) (Table 2). In addition, the model was evaluated by determination coefficient R^2^ = 99.52%, indicating that 0.48% of total experiments were not explained by the model. The R^2^ adjustment and R^2^ prediction for EPS production were 0.9908 and 0.9755, respectively, which designated good results with a difference less than 0.2. The measurement by the signal-to-noise model, assessed by adequate precision, was acceptable, with a value of more than 4 in this model, being 58.29. From Table 2, it can be noticed that all terms for EPS production exhibited a significant effect (*p* < 0.0001) except the D^2^. C-pH was found as the most effective parameter with the highest F-value (539.58). Here, the significance parameters were evaluated by F-value as all *p*-values showed as <0.0001 [28].

Furthermore, a regression optimization model was constructed, showing a relationship between independent parameters and dependent parameters for EPS production. The model was developed based on ANOVA and the regression coefficient. The CCD matrix was fitted in a quadratic optimization model in coded parameters, as described in Equation (8).
EPS = 4.0816 − 0.423 * A − 0.623 * B − 0.710 * C + 0.3541 * D + 0.52 * A * B + 0.57 * A * C + 0.24 * A * D + 0.64 * B * C + 0.17 * B * D + 0.069 * C * D − 0.58 * A2 − 0.47 * B2 − 0.72 * C2 + 0.034 * D(8)

In addition to ANOVA and the correlation optimization model, regression analysis was applied to evaluate the best-fit line among the experiments that can be visualized by the normal plot of the residuals (Figure 2A). The observed experimental values plotted against predicted output values for EPS production are shown in Figure 2B and exhibit good aggregation between the observed and predictive values estimated by the quadratic optimization model.

Contour plots (2-D) are a graphical representation that help to visualize the actual state of the regression optimization model to infer the relationship between parameters and response. Circular or elliptical curves indicate that corresponding parameters have not interacted, while hyperbolic and plateau-shaped curves denote interaction. In this study, the relationship between independent parameters and EPS production was explained by a 2-D contour plot of the response surface, while keeping others at zero level (coded value) [28]. In Figure 3a, it can be observed that the maximum yield of EPS was 4.6 g L^−1^ when the concentration of glucose and yeast extract were both 20 g L^−1^, while the minimum EPS yield was 0.1 g L^−1^ with high glucose and yeast extract concentrations (50 g L^−1^). Figure 3b illustrates glucose interaction with pH on EPS production, displaying a maximum EPS production of 4.5 g L^−1^ (glucose 20 g L^−1^, pH 6.0), with the minimum yield being produced when the glucose concentration and pH value increased. From Figure 3c, it can be seen that EPS production becomes highly affected by an increase in the glucose concentration to 50 g L^−1^ with an inoculum size of 10–30 mL L^−1^. Figure 3d explains the interaction between yeast extract and pH, and reveals a high EPS production (4.8 g L^−1^) when the pH was 6.0 and the yeast extract was 20 g L^−1^. A low yield was observed at a high yeast extract concentration and an alkaline pH. Interestingly, the interaction of the nitrogen source with inoculum size in Figure 3e demonstrates the same nature as that observed in Figure 3c between the carbon source and the inoculum size. Finally, Figure 3f displays the interaction between pH and inoculum size on EPS yield, where yield increased with a slightly acidic pH 6.0 and an inoculum size of 50 ml L^−1^.

Based on the optimization model, the ramp chart for optimization conditions was developed by Design Expert software (version 7), as displayed in Appendix A. To validate the optimization result and model accuracy, the experiment was carried out in the laboratory with the suggested optimum conditions of glucose (20 g L^−1^), yeast extract (20 g L^−1^), pH 6.00, and inoculum size (20 mL L^−1^) in triplicate. The result revealed an actual EPS yield of 6.05 g L^−1^, which is a good aggregate of the predicted result (5.98 g L^−1^). The RSM optimization study resulted in an almost two-fold increase in EPS production. The optimized high yield EPS was purified and used for further characterization experiments.

### 3.5. Physicochemical Characterization of the EPS

#### 3.5.1. Morphological Analysis (AFM and SEM)

AFM and SEM were used, respectively, to understand the surface morphology and three-dimensional structure of the extracted EPS. AFM is an extensively used powerful tool to study morphological characteristics by measuring the interaction forces in liquids at a pico-Newton or nano-Newton level with a high lateral or vertical resolution [40]. As observed from SEM analysis at 3000×, 5000×, and 10,000× magnifications, respectively, the EPS biopolymer has a highly compact, non-porous, flake-like compact structure, as previously described by Chen et al. [41]. It exhibited amorphous surface roughness composed of evenly distributed macromolecular lumps (Figure 4A–C), whereas the topographical two-dimensional (2D) AFM image confirms that these macromolecular lumps are spherical and are of different sizes (Figure 4D). The three-dimensional (3D) AFM topographical image (Figure 4E) indicates that these irregular-sized, spherical lumps have spike-like heights, ranging between ~1.5 nm and ~3 nm (Figure 4F). 

SEM-EDS analysis of EPS produced by CamB6 revealed that the biopolymer is mostly composed of carbon (% total weight 52.52) and oxygen (% total weight 47.48). The EPS was found to have traces (<0.5% *w*/*w*) of nitrogen, sulfur, and phosphorous elements.

#### 3.5.2. Monosaccharide Composition Analysis and Molecular Weight

Sugar analysis performed using the HPLC method showed that the EPS produced by *B*. *haynesii* CamB6 is a heteropolymer, chemically composed of mannose (66%), glucose (20%), and galactose (14%) monosaccharides. The HPLC chromatogram is provided in Figure 5A. 

In Figure 5B the GPC chromatogram of CamB6 EPS is presented and shows a broad bell-shaped peak. From this GPC analysis, it is confirmed that the EPS produced by *B. haynesii* CamB6 has an average molecular weight (Mw) of 53.6 kDa. 

#### 3.5.3. Structural Analysis 

##### Infrared Spectroscopy

FTIR spectroscopy was used to determine the main functional groups and the chemical structure of the EPS. The FTIR spectra of *B. haynesii* EPS are shown in Figure 6. 

The spectrum showed a broad band centered at 3302 cm^−1^, which corresponds to the sugar residue ν–OH groups stretching vibration (-H bonded). The band at 2926 cm^−1^ belongs to the (ν-C-H) stretching vibration in the sugar ring [42]. The absorption band at 1642 cm^−1^ is due to the stretching vibration (ν C=O, Amide I) [43]. The Amide I band and that at 1725 cm^−1^ (ν C=O, COOH/-COCH_3_), and 1547 cm^−1^ (ν-C-N (C-N-H) + δNH, Amide II), suggest the presence of protein traces in the EPS [44], as previously revealed by the elemental analysis. The 1725 cm^−1^ band could also suggest the presence of acetyl (-COCH_3_) groups in the EPS monomeric units. The peaks at 1421 cm^−1^ (δ-CH_2_) and 1373 (δ-CH + δC-CH_3_) are due to C-H groups. Between 1200 and 950 cm^−1^, a group of intense overlapped bands appears in a spectral region known as the “sugar region”, where stretching vibrations of glycosidic bonds and pyranosic rings predominate [45]. This is the case of the bands at 1208 and 1126 cm−1 (ν^as^ C-O-C, ring), and 1024 and 969 cm^−1^ (ν C-O, C-C) in the α-ring, respectively. New bands between 950 and 750 cm^–1^ appear in the wavenumber range known as the “anomeric region”. In this region, the ring anomeric bands together with other complex skeletal vibrations are included. For instance, the presence of an α-anomeric configuration of mannose sugar units was suggested by the sharp band at 809 cm^−1^ (γ_CH_ C_1_ axial of α- linkage), whereas the presence of a β-anomeric configuration was suggested by the absorption at 882 cm^−1^ (δ_CH_, C_1_ axial of β-linkage) [46,47]. This result indicates that there could be more than one type of linkage in the EPS backbone or branching. 

##### 1D and 2D NMR Analysis

To further characterize the EPS, 1D (1H and 13C) and 2D NMR (1H [obsd.] 13C, COSY, HSQC) was carried out to propose a partial EPS structure. Some of the 1D and 2D spectra are shown in Figure 7.

Firstly, our tentative assignment of proton signals is based on previous work on polysaccharides and sugar analysis [46,48,49]. In the low field region (δ5.4-4.3), 1H NMR spectra (Figure 7A) of the EPS show different signals corresponding to the protons from anomeric carbons of sugars, suggesting different types of monosaccharides linkages (α and β) in the EPS structure, as previously described by the FTIR study. The complexity of signal multiplicity may be due to the difference in the chemical environment and substitution (type of linked sugars) for reducing and non-reducing end units. The anomeric proton signals at δ5.34 and δ5.12 ppm belong to α-mannobiose (1→4) and α-mannobiose (1→2), respectively. These signals agree with those reported for an α-D-Manp-(1→4) EPS isolated from *Aspergillus* sp. Y16 strain [46]. On the other hand, the 13C NMR spectrum shown in Figure 7B contains several signals in the C-anomeric region. The EPS structure contains five C-1 corresponding to the mixture of carbohydrates D-Glcp, D-Manp and D-Galp as determined by HPLC analysis. The C-1 signal of α-D-Manp δ100.8 corresponds to 2,4-di-O-substituted units. We assign the linkage of α-D-Manp-(1→4)-α-D-Manp in concordance with the chemical shifts previously reported for the galactomannans from the lichen *Rocollela decipiens* Darb [46,50]. Besides, the α-D-Manp-(1→2)-α-D-Manp was attributed to δ102.3, which also appears in α-mannotriose. Moreover, the heteronuclear 1H-13C HSQC spectrum of EPS displays the α-mannobiose (1→2) δ 5.34, 100.8 for α-D-Manp-(1→2)-α-D-Manp and α-mannobiose (1→4) δ 5.12, 102.3 for α-D-Manp-(1→4)-α-D-Manp [50], as can be seen in (Figure 7C) Figure 1.

In the COSY spectrum (Figure 7D) its α-configuration by low-field H-1 signal at δ5.34 was found and correlated with H-2 (δ4.0), H-6 (δ3.98 and 3.85), and H-5 (δ4.12) [49]. The β-glucopyranose units appear at δ4.83 ppm [51]. Besides, the anomeric protons from β-galactopyranose units appear at δ4.63 and 4.47 ppm. Finally, an acetyl (–CH_3_) signal at δ2.00 and 2.04 ppm suggests EPS monosaccharide acetylation. This is a confirmation of results obtained from the FTIR analysis.

#### 3.5.4. Thermogravimetric Analysis

Thermal analysis has become a widely used technique for the thermal stability analysis of polymers. In Figure 8, thermal decomposition curves (TG and DTG) of EPS are presented. 

The existence of three decomposition peaks in the thermogravimetric curves suggests that thermal degradation of EPS is a complex process. This multistep feature is consistent with the heteropolymeric nature of this EPS, composed of three different sugar residues, as previously described. The first degradation step occurs from 25 to 107 °C, showing a maximum decomposition rate temperature (T_Peak_) at 56.3 °C, accounting for 4.5% of weight loss. This process is mainly associated with dehydration due to moisture absorption of EPS. A second and more significant effect occurs from 200 to 279 °C, peaking at 240.7 °C, showing a weight loss of 36.5%. This is the effect with the highest mass loss during EPS thermal degradation. In this step, depolymerization of polysaccharide and protein chains, and the thermal scission of chemical bonds take place, accompanied by dehydration of sugar units [4]. The third effect during EPS decomposition occurs between 279 and 380 °C, peaking at 310 °C with an associated weight loss of 16%.

### 3.6. Functional Properties of the EPS

#### 3.6.1. In Vitro Antioxidant Activity Determination

Here, in vitro antioxidant activities of the EPS were assayed by their scavenging ability against DPPH, H_2_O_2,_ and ABTS compared to standard ascorbic acid (Figure 9A–C). EPS produced by CamB6 showed significant free-radical scavenging activity in comparison to the standard ascorbic acid. At low concentration, i.e., 0.5 mg mL^−1^, good ABTS-mediated free-radical scavenging activity of more than 60% was shown. In the case of all three experiments, the highest tested concentration showed good % radical scavenging activity, which is near to the ascorbic acid. It ranged from 70% to 80% in the case of all three tested free-radical scavenging activities. The maximum activity achieved by DPPH, H_2_O_2_, and ABTS-mediated methods were, respectively, 72.34%, 76.21%, and 72.80%.

#### 3.6.2. Emulsifying Activity Study

In this study, the emulsification activity of the EPS produced by CamB6 has been compared to commercial chemical surfactant, Tween 20 (Merck), and a commercially used bacterial biopolymer, Xanthan gum (Merck). All the studies were performed in triplicate. In our study, EPS was found to be capable of stabilizing different vegetable oils by creating a hydrophobic phase (Table 3). The CamB6-produced EPS was found to be more efficient than the chemical surfactant used as a control for all the studied vegetable oils. Emulsification activity was found to be more efficient in the case of the studied EPS than in the control commercial bacterial EPS xanthan gum, except in the case of olive oil and sesame oil. Our study EPS exhibited stable emulsion formation against five different food-grade vegetable oils (corn 69.09 ± 0.25, avocado 69.09 ± 0.25, canola 64.55 ± 1.20, coconut 62.73 ± 0.97, and sunflower 62.73 ± 0.65). 

#### 3.6.3. Flocculation Activity Study

The flocculation activity of the study EPS was analyzed using kaolin clay suspension (Figure 9D). The percentage flocculation of the EPS for kaolin suspension demonstrated a gradual yet sharp increase to 60 mg L^−1^ concentration, which showed a little decrease with the further increase in the EPS concentrations. The highest % flocculation of 52.3 was observed with 60 mg L^−1^ EPS concentration, while the lowest was 36.9% (100 mg L^−1^).

### 3.7. Water-Holding and Oil-Holding Capacity Determination

From the obtained results, the study EPS showed 102.9% water retention (WHC). According to the statistical analysis using ANOVA, there are significant statistical differences (*p* < 0.05) associated with the control xanthan gum, which presented greater stability and affinity with water, delivering a value of 183.3%. OHC is a vital property of the EPS and is associated with the permeable structure of the carbohydrate polymer chains. It was observed that the OHC of the study EPS showed a better response and statistically significant difference of 111.9%, compared with that of the commercial xanthan gum with 111.0%. 

## 4. Discussion

Numerous works document extremophilic microbial community analysis from Chile, such as the Chilean Altiplano in the north [6,52] or Chilean Patagonia in the South [53,54,55]. To our knowledge, our study is the first report of any structurally and functionally characterized EPS from Chilean extreme environments. Here, we have specifically focused on a functional EPS produced by thermophilic *B. haynesii* CamB6 from an Andean hot spring for food applications. As the study site is situated in the Laguna del Maule stratovolcano field, the geothermal water of Campanario is reported to have chemical signatures of fluids from both local magmatic reservoirs and meteoric water circulation [56]. The high SO_4_-Cl (sulfate 207.5 ± 8.7 mg L^−1^, chloride 12.0 ± 0.3 g L^−1^) content of the water collected from Campanario confirms its “acid spring” nature, implying that the sample water is deep volcanic thermal water. Along with chlorides, the presence of fluorides (F- 1.2 mg L^−1^) also indicates the presence of magmatic volatiles in the study water [57]. In addition, the presence of chloride also clearly hints at the presence of a volcanic arc beneath the hot spring [58]. Apart from that, as may be observed from Appendix A, the study site is also rich in different metals and heavy metals (Al, Cu, Mg, Mn, Ni, Pb, Zn, As). The metal deposits are mainly from the volcanic and volcanoclastic rocks formed during quaternary volcanism. These are also responsible for augmenting the salinity of the water due to their interactions with sedimentary rocks [59]. Thus, the microbiota inhabiting these thermal fluids are exposed to multiple extreme factors, viz., temperature, acidic pH, UV rays, salinity, and the presence of heavy metals. Over the last decade, polyextremophilic ecosystems were often explored in a bid to understand their life forms and also to fulfill their biotechnological interests, as extremophiles are known to produce value-added bioactive compounds as a part of their stress resistance. The southern hemisphere, particularly Chile, has always been used as a model to understand the extremophilic ecosystems [52,60]. In central Chile, specifically, however, the Maule region hot springs are much less reported in terms of their bacterial life forms and bacterial bioactive compounds, one of which is the focus of our present study.

The thermophilic bacterial isolate CamB6, studied in this manuscript, optimally produced 5.6 g L^−1^ EPS at 55 °C in a 2% glucose (carbon source) and 2% yeast extract (nitrogen source) supplemented medium. According to previous reports, it should be noted that thermophilic *B*. *thermoruber* 423 isolated from Bulgarian hot springs was reported to have the highest EPS yield of 0.897 g L^−1^ [61]. Thermophilic *Geobacillus tepidamans* isolated from Velingrad hot spring, Bulgaria, had a yield of 0.114 g L^−1^ EPS [62], and a thermophilic, endospore-forming isolate from the radon in the hot spring had a yield of 0.06 g L^−1^ EPS [44]. All of these are much lower compared with our study (~6 g/L). Therefore, our study EPS can be considered a high yield. The developed optimization model can be generalized for EPS production by *B. haynesii* CamB6 under the same conditions. From the ramp chart (Appendix A), it can be seen that the desirability score of the optimization model is 0.984. Our study is the first report to characterize the EPS production by *B. haynesii*. However, the goodness of fit of the model by examining the determination coefficient (R^2^) was documented earlier for different *Bacillus* species. For *B*. *subtilis*, R^2^ was 0.948 [63], *B*. *circulans* R^2^ was 0.9614 [64], *B*. *velezensis* R^2^ was 0.990 [65], *B*. *mycoides*, R^2^ was 0.976 [66], etc. Thus, comparing the desirability of the model of different other *Bacillus* isolates, our study species shows a good aggregate of the result. In our case, the best conditions were used to prepare EPS in a sufficient quantity to allow further purification and characterization by different physicochemical techniques.

From the morphological analyses of the study EPS, an amorphous, non-porous, irregular, and dense surface denotes the compact nature of the biopolymeric material. This characteristic also confirms the film-like feature of the biopolymer [67]. It indicates a pseudoplastic behavior due to the strong interaction between water molecules and the hydroxyl groups (−OH) of the EPS, which is in line with the reports by Wang et al. [29] and Kanamarlapudi and Muddada [68]. In both reports, the compact structure of the EPS is interpreted as being useful for application in the food industry, most likely as a potential food additive [29]. The macromolecular lumps observed on the compact surface could be formed due to intra and inter-molecular aggregation of macromolecules, as mentioned previously for other EPSs [39]. From both AFM and SEM studies, EPS was observed to exhibit a plasticized, film-like, compact nature, which may result in exceptional thickening, as well as viscosifying properties with potential applications as biotechnological additives [69]. The sugar analysis confirmed that the EPS was composed of mannose, followed by glucose and galactose units in a 3.3:1.0:0.7 ratio. Bacterial EPS produced by hot-spring-origin thermophilic *Geobacillus* sp. was already noted earlier as being mainly composed of these three monosaccharides in a different ratio [70]. Other *Bacillus* species are reported with similar monosaccharide constituents with different molar ratios, making each one unique [71,72]. As mentioned earlier, from the FTIR analysis, both α- and β-anomeric configurations were suggested as being part of the EPS structure. This result indicates that there could be more than one type of linkage in the EPS backbone or branching. Thus, a further detailed linkage analysis could be performed in future to clarify this point. Also, the presence of acetyl groups was determined. However, the exact location of the acetyl group position would be a matter for further work. The results of NMR analysis also suggest that *B*. *haynesii* EPS has a complex chemical structure and confirm FTIR results. The EPS structure contains five C-1 corresponding to the mixture of carbohydrates D-Glcp, D-Manp and D-Galp, as also determined by HPLC analysis. It is a heteropolysaccharide probably formed by an α-mannobiose (1→4) and α-mannobiose (1→2) backbone with different branches of neutral sugars (Glc and Gal) attached by different glycosidic linkages (α and β), where some sugar units are also acetylated. Regarding the EPS thermal stability, the thermal decomposition of the EPS occured in three steps, the first one being associated with water loss. The second step can be explained by the relatively higher content of glucose and mannose units in the EPS; it is expected that glucomannan-type fractions could thermally decompose at this temperature. Recently, some authors reported that the thermal degradation of konjac glucomannan, a polysaccharide formed by β-1,4 linked D-glucose and D-mannose units, also proceeds in two thermal steps [4]. The first was associated with water loss, and the second showed a maximum decomposition rate temperature T_peak_ of 289 °C. In this step, the depolymerization of polysaccharide and proteins chains and the thermal scission of chemical bonds occurred, accompanied by dehydration of sugar units. This temperature was 10 °C higher than that observed in the EPS but this fact could be associated with the type of sugar linkage in glucomannan (β-1,4), which provides high thermal stability. Moreover, the thermal stability of *B*. *haynesii* EPS is similar to that reported for EPS1 from *B. licheniformis* T14, which started to decompose at 210 °C and reached a peak at 260 °C [73]. The third step of the thermal stability of this EPS is similar to the degradation temperature reported for galactomannans isolated from three different plant seeds [74]. In that study, the weight loss temperature started between 285 and 297 °C, the T_peak_ ranged from 31 to 321 °C, and the galactomannans had Man/Gal ratios of 4.21:1, 2.55:1, and 3.03:1, respectively. The proposed structure suggests a β-D-mannopyranose backbone with galactose side chains at the C_6_ position. The *B*. *haynesii* EPS from the present work has a Man/Gal ratio of 4.9:1. Taking into account the peaking temperature, the thermal decomposition range, and the lower mass losses associated with it, this thermal effect seems to be associated with the galactomannan unit decomposition. In summary, *B*. *haynesii* EPS shows similar thermal stability to other commercially available plant polysaccharides. As food industrial applications often require a wide range of temperature adaptability, the good thermal stability of the study EPS encourages us to study potential applications of this EPS for food application. 

The presently studied EPS from thermophilic bacteria has ropy characteristics which help in the formation of good texture and can be used in the food industry, as previously investigated by Marshall and Rawson [75]; they found that thermophilic EPS-producing lactic acid bacteria could be used to produce in good-textured yogurt. On the other hand, extremophilic exopolysaccharides are comparatively non-pathogenic in nature, making them suitable for application as emulsifiers, gelling agents, suspending agents, and stabilizers, etc., in the food industry [76]. As per one example [62], gellan, a microbial heteropolysaccharide from *Geobacillus stearothermophilus,* can be used as a gelling substance and suspending agent in the food industry. Thus, in this study, in vitro activities of the EPS were checked to determine its efficacy. 

Regarding the functional bioactivity (antioxidant properties DPPH, ABTS, and H_2_O_2_), the *B. haynesii* EPS showed a concentration-dependent effect, which is similar to the previously reported results for polysaccharides from four Auriculariales [77]. In the case of the ABTS and DPPH methods, dose-dependent antioxidant activity was observed with 60–65% scavenging at 0.5–1.0 mg mL^−1^ EPS concentrations. The H_2_O_2_-mediated antioxidant activity showed dose-dependent effects similar to the result obtained in the case of DPPH (Figure 7A). Depending on the previously mentioned EPS concentration values, around 60% of radical scavenging percentage, all methods have shown a stationary curve with an increasing concentration of EPS. The higher antioxidant activity (76–78%) was achieved in all cases at 5 mg mL^−1^ of EPS concentration. This good microbial EPS-mediated antioxidant activity was also reported previously for other neutral EPSs [78,79]. It has been reported that the antioxidant properties of EPSs are associated with the sugar composition of polysaccharides and their physicochemical properties [77]. EPSs in thermophilic bacteria create an environment that protects cells against free radicals produced by high temperature [80]. Antioxidants play an important role against various diseases and normal aging processes owing to free radical scavenging properties [46]. In addition, antioxidant capacity is also an interesting property for improving food shelf life because antioxidant compounds inhibit the oxidation of vitamins and other nutrients during food storage [81]. The in vitro antioxidant activity shows prodigious potentiality to be developed as an antioxidant agent, as well as a functional additive for the food industry. A similar study was previously undertaken by Gongi et al. [82], in which thermophilic *Gloeocapsa gelatinosa* showed similar results. Thus, EPS with an improved antioxidant capacity is reported to be utilized as antioxidant food additives [2].

From the results of the emulsification activity, it can be said that our study EPS can efficiently emulsify the fatty acids present in the chemical structure of the food-grade vegetable oils. It is interesting to note that the emulsification capacity of the EPS was similar to, and sometimes higher than, that of the commercial EPS biopolymer, xanthan gum. It is known that due to the spatial stability of EPSs, an extensive network can be formed in the continuous phase to stabilize the emulsion [83]. The emulsions have numerous applications in food industries, such as dairy, candy, beverage preparation, meat, and milk processing [80]. This result indicates that the EPS can be used as a potent emulsifier and stabilizer in the food industry as it can develop and stabilize both oil-in-water, and water-in-oil emulsions [84]. The EPS-stabilized emulsions are particularly interesting for food applications due to their thickening properties, as they develop a macromolecular block in the aqueous medium between the dispersed droplets [84].

On the other hand, the EPS showed stable flocculation activity. Generally, the effectiveness of biopolymer flocculation depends on charge density and the presence of charged function groups [85]. At a low EPS concentration, there may be a low % flocculation, as *B*. *haynesii* EPS is a neutral macromolecule and some charged groups on it could be associated with remaining protein traces. In the case of more than the optimum concentration, the percentage of flocculation decrease may be due to the generation of high-viscosity solution blocking the adsorption sites [86,87]. A similar trend of a decrease in the percentage of flocculation at a higher EPS concentration was reported in recent work [87]. Flocculation is a desirable property of industrial food fermentation processes, allowing the easy separation of cells from the product [88]. Polysaccharides are widely used as a flocculent aid in food processing [89].

WHC is one of the functional properties of polysaccharides that are highly influenced by factors, such as molecular weight, particle size, ionic forms, and the composition of polysaccharides [90]. Previously, *B*. *licheniformis* EPS showed 98.8% WHC [91], which is more compared with another earlier report on Lactobacillus EPS which only showed 8.95% WHC [92]. Our study EPS produced by hot-spring-origin *B*. *haynesii* CamB6 demonstrated better activity (102.9%) in comparison with both these reports. However, the animal-origin polysaccharide, chitosan, obtained from crab (138%) and shrimp (358%), is observed to be significantly higher than all these reported bacterial EPSs [36]. The reason behind the low WHC of our study EPS could be due to its low molecular weight and less porous nature. According to Levine and Slade [93], EPSs with a lower %WHC can be useful to improve crispness and improve workability for chips, crackers, or snack preparation. This indicates that our study EPS might have a future application as an additive or texture enhancer for low-moisture baked foods. The OHC in our study is much higher than the reported OHC for EPS produced by *Weissella confusa* with 5.1% [94] and *Lactobacillus* EPS with 15.9% [92]. However, it is similarly lower than chitosan with 635% [95]. OHC is one of the important characteristics of polysaccharides, as it indicates the adsorption of organic compounds or oils to the surface of substrates. Thus, EPSs with significant OHC may be used as additives in different food industry applications, such as sausage preparation [96]. Although chemical composition plays an important role in the OHC, the porosity and the affinity of the biopolymer with the oil also contribute to the activity. The EPS produced by *B*. *haynesii* CamB6 presents good results with the possibility of future usage in different food industry processes.

Our study EPS shows promising in vitro free radical scavenging and emulsification properties against vegetable oils at low concentrations, while the standard flocculation property in a moderately high concentration denotes its different food technological functions. Considering the applications of the EPS produced by thermophilic *B. haynesii* CamB6 as an antioxidant, emulsificant, and bioflocculant, it is important to consider the biotechnological potential of the earth’s extreme environments. As mentioned earlier, Chile is a natural laboratory to study microbial community dynamics in polyextreme ecosystems; this study is a first report of any bacterial EPS from polyextreme ecosystems in Chilean hot springs. Our present study is an approach to displaying the potential of the Chilean polyextreme environment and thermophilic *Bacillus* species for food applications.

## 5. Conclusions

The findings of this study on EPS produced by thermophilic *B. haynesii* CamB6 isolated from a relatively unexplored Chilean hot spring culminates in the use of extremophilic bacterial EPS in different food technological applications, such as an antioxidant, emulsifier, or flocculating agent. Bacteria produce EPS as a protecting sheath against extreme environmental conditions. Chile, having a natural ecosystem of different polyextreme environments, harbors immense biotic potential in addition to EPS production. Though microbial communities in different Chilean hot springs were already reported, to our knowledge, this study is the first to highlight a functional EPS from any Chilean hot spring. Alongside, it is also the first to elucidate the EPS produced by *B. haynesii,* both structurally and functionally, for different food applications.

## Data Availability

Data are contained within the article and Appendix A.

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
