# Peer review of "Optimization and Characterization of a Novel Exopolysaccharide from Bacillus haynesii CamB6 for Food Applications"

_biomolecules, 2022, doi:10.3390/biom12060834_

Round 1
Reviewer 1 Report
See file

Author Response
Please see the attachment to see the responses.

Reviewer 2 Report
The presented manuscript is related to the study of an exopolysaccharide isolated from the extremophilic microorganism Bacillus haynesii CamB6. The authors used the RSM based on central composite design technique to maximize the EPS production by B. haynesii, and also conducted a diverse study of the EPS properties. The studied EPS showed promising properties to use in the food industry. In combination with a high yield of polysaccharide from the bacterial mass, this makes it attractive for practical use.
Basically, the work was done at a fairly high professional level, the material is presented clearly, well structured, the data obtained are discussed with the previously obtained literature data. Methods are described adequately.
Major remarks
- To my deep regret the structural characteristic of EPS is the weak point of this publication. Fourier transform infrared spectroscopy (FTIR) and NMR spectroscopy data can only be regarded as preliminary. According to the 1H NMR spectrum, it can be assumed that EPS has a complex branched structure. Probably, to establish its structure, it is necessary to use chemical methods (hydrolysis, methylation, etc.), as well as two-dimensional NMR spectroscopy.
- EPC is isolated from a thermal spring of volcanic origin, which contains a set of various elements, including heavy metals. It is advisable, in my opinion, to provide information on the content of heavy metals in the EPS, since the authors plan to offer this product for the food industry in the future.
Minor remarks:
Line 42: It is premature to include the term "structure" in keywords at this stage.
Line 526-527: The proposal should be reformulated. According to Figure 7. the activity of the ascorbic acid is higher than the activity of EPS produced by CamB6.
Round 2
Reviewer 1 Report
The English language was improved, but there are still errors.
The following comments were not addressed correctly:
Why was linkage analysis not performed? (I was referring to the chemical method by methylation.)
Bacterial EPSs are normally composed of repeating units. Based on the 1H NMR spectrum, the purity and/or heterogeneity of the EPS is questioned. (I was referring to peak intensities that are not uniform. The new spectra emphasize this even more: for example, I count at least 10 H1/H2 cross peaks of different intensities on the COSY.)
The authors added a COSY spectrum but did not analyze it. They deleted erroneous information about mannose residues, but came up with new conclusions about mannose identity and linkages based solely on chemical shift of anomeric proton/carbon.
TOCSY and HMBC are mentioned in methods, but no results are presented. The information contained in the other 2D spectra was not exploited to provide structural information. Chemical shift similarity is not sufficient to identify monosaccharide and linkage in a polysaccharide repeating unit. The evidences for the partial structure given are not convincing. If the authors do not add methylation analysis and perform full NMR spectral analysis, the claims about structure determination should be removed from the manuscript.
Reviewer 2 Report
The authors took into account the comments of the reviewer and significantly improved the article.
Author Response
Thank you so much dear reviewer for your positive comments regarding the revised version of our manuscript.